# Campylobacter Species of the Oral Microbiota as Prognostic Factor for Cardiovascular Outcome after Coronary Artery Bypass Grafting Surgery

**DOI:** 10.3390/biomedicines10081801

**Published:** 2022-07-27

**Authors:** Susanne Schulz, Britt Hofmann, Julia Grollmitz, Lisa Friebe, Michael Kohnert, Hans-Günter Schaller, Stefan Reichert

**Affiliations:** 1Department of Operative Dentistry and Periodontology, Martin-Luther-University Halle-Wittenberg, 06112 Halle, Germany; julia.grollmitz@uk-halle.de (J.G.); lisa.friebe@hotmail.de (L.F.); michael_kohnert@gmx.de (M.K.); hans.guenter.schaller@uk-halle.de (H.-G.S.); stefan.reichert@uk-halle.de (S.R.); 2Department of Cardiothoracic Surgery, Heart Centre of the University Clinics Halle (Saale), Martin-Luther-University Halle-Wittenberg, 06112 Halle, Germany; britt.hofmann@uk-halle.de

**Keywords:** oral microbiota, high-throughput sequencing, *Campylobacter* species, cardiovascular disease, adverse cardiovascular events, longitudinal cohort study, 3-year follow-up

## Abstract

Background: The oral microbiota has been implicated in a variety of systemic diseases, including cardiovascular (CV) disease. The main objective of this study (DRKS-ID: DRKS00015776) was to evaluate the prognostic importance of the oral microbiota for further CV events in patients undergoing coronary artery bypass grafting surgery (3-year follow-up). Methods: In this longitudinal cohort study, 102 CV patients were enrolled, of whom 95 completed the 3-year follow-up. The CV outcome was assessed using the major adverse cardiac and cerebrovascular events criteria. To evaluate subgingival colonization, 16S rRNA genes were amplified, targeting the V3/V4 region (Illumina MiSeq). Results: Regarding the specific number of operational taxonomic units (OTUs), no significant differences in CV outcome were determined (alpha diversity, Shannon index). In linear discriminant analyses and *t*-tests, the disease-specific differences in the beta diversity of the microbiota composition were evaluated. It was evident that bacteria species of the genus *Campylobacter* were significantly more prevalent in patients with a secondary CV event (*p* = 0.015). This hierarchical order also includes *Campylobacter rectus*, which is considered to be of comprehensive importance in both periodontal and CV diseases. Conclusions: Here, we proved that subgingival occurrence of *Campylobacter* species has prognostic relevance for cardiovascular outcomes in CV patients undergoing coronary artery bypass grafting.

## 1. Introduction

For many years, cardiovascular (CV) diseases have been the number one cause of death worldwide [1,2], in Europe [3], and, especially, in Germany [4]. In 2020, diseases of the circulatory system topped the list of causes of death in Germany at 34.3% [4]. As reported by the Federal Statistical Office (Destatis) in 2017, CV diseases accounted for the highest percentage of total healthcare costs, at around 13.7% [5]. Due to the high morbidity and mortality of CV diseases and the associated social and financial impact, the focus of clinical research is on the prevention, etiology, pathogenesis, and therapy of the disease.

The etiology of CV diseases is characterized by multifactorial impacts that include a variety of clinical risk factors (e.g., elevated blood pressure and diabetes), environmental risk factors (e.g., air pollution), and health behavior risk factors (e.g., smoking, dietary factors, and physical inactivity) [3]. In addition to already established CV risk factors, clinical research is focusing on other possible mediators that influence CV disease [3,6].

In this respect, periodontitis, as a microbiota-induced inflammatory disease, has been implicated as a risk modulator of CV disease [7,8,9,10].

There is epidemiological evidence for a bidirectional association between CV disease and periodontitis: CV patients show a higher incidence of periodontitis, and the occurrence of CV events is increased in patients with periodontitis [9]. Many mechanisms, based on biological plausibility, could explain the epidemiological link between CV disease and periodontitis. In this context, the risk of bacteremia and/or endotoxemia, which can be caused by everyday activities such as brushing teeth, and by professional dental intervention as well, has also been discussed [11]. Indeed, antigens derived from oral bacteria and viable periodontal pathogens (*Aggreagtibacter actinomycetemcomitans, Porphyromonas gingivalis*) have been detected in atherothrombotic tissues [12,13,14]. This strengthens the assumption that oral infection impacts on CV disease [15]. In intervention studies, periodontitis treatment has been proven to help prevent or delay progression of CV diseases [16,17]. In addition to the primary prevention of CV diseases, a major focus has also been placed on preventing further CV events (secondary prevention). Indeed, a pilot study showed that patients with CV disease are less likely to experience a serious CV event over a 25-month period when treated with scaling and root planing [18].

In this respect, other factors contributing to periodontitis and their possible impact on CV secondary events have also been discussed. Recurrent oral inflammation due to dysbiosis of the oral microbiome is also considered to play an important role as a CV prognostic factor [19]. In line with this hypothesis, periodontal pathogenic bacteria and their corresponding antibodies were evaluated for their potential prognostic relevance in CV diseases, but the results were contradictory [20,21,22,23,24]. Notably, only individual bacterial species or antibodies were investigated in previous studies and, according to the polymicrobial synergy and dysbiosis model, the oral microbiome acts as a complex and is associated with innate and adaptive host immune responses in order to trigger periodontal and systemic infection [25,26,27]. For this reason, complex consideration of the oral microbiota was the focus of the present longitudinal cohort study in analyzing CV prognosis. In the 1-year follow-up, we demonstrated that oral bacteria were associated with CV prognosis according to major adverse cardiac and cerebrovascular events criteria (MACCE) [28]. In a continuation of this study, here, we aimed to verify in a follow-up period of 3 years that the oral microbiota also plays an important role in postoperative outcomes after coronary artery bypass grafting (CABG) surgery. Possible differences in the oral microbiota that could enable predictions to be made about CV disease progression could help to better assess CV prognosis. In this case, evaluating the oral microbiota as part of an individualized therapy may offer the potential to prevent secondary CV events. In addition, the results of this study may provide a platform for targeted intervention studies.

## 2. Materials and Methods

### 2.1. Study Design

The present work was registered in the German Clinical Trial Register (Oral microbiome and transcriptome as predictors of new CV events in patients with CHD and need for cardiac surgery; DRKS-ID: DRKS00015776). This study was approved by the ethics committee of the Martin-Luther University Halle-Wittenberg (registration no. 2016-86). Informed written consent was obtained from each patient. All investigations were carried out in accordance with the ethical guidelines of the Declaration of Helsinki and its amendment in Tokyo and Venice. This study was conducted as a longitudinal cohort study with a 3-year follow-up period.

### 2.2. CV Patients

#### 2.2.1. Baseline Investigations

Patient characteristics and investigations were described in detail in Reichert et al., 2021 [29]. Briefly, between January and October 2017, 308 CV patients undergoing CABG surgery were screened using the inclusion criteria and, ultimately, 102 patients were consecutively recruited at the Department of Cardiothoracic Surgery of the Heart Centre of the University Clinics Halle (Saale). The inclusion criteria were as follows: age > 18 years, number of teeth > 4, at least 60% stenosis of one of the main coronary arteries demonstrated by angiography, and CABG surgery mandatory. The exclusion criteria were pregnancy, subgingival scaling and root planing and/or antibiotic therapy during the last 6 months, requirement of endocarditis prophylaxis (according to the German Society for Cardiology [30]), inability to give written informed consent, current alcohol or drug abuse, and intake of drugs that potentially cause gingival hyperplasia.

Baseline variables (age, gender, smoking habits, and body mass index) and patients’ medical history (e.g., diabetes mellitus, hypertension, peripheral arterial disease, and dyslipoproteinemia), serum parameters (INR (International Normalized Ratio) score, hemoglobin (Hb), hematocrit, creatinine, urea, glycated hemoglobin (HbA1c), C-reactive protein (CRP), leukocytes, platelets), and medications of the CV patients (lipid-lowering drugs, oral anticoagulants, and antiarrhythmics) were assessed.

The dental examination was performed on the day before CABG surgery. This evaluation comprised determining the plaque index (PI; four tooth surfaces) [31], maximum clinical attachment loss (CAL = distance between the cementoenamel junction and the bottom of the pocket, six points around each tooth), clinical probing depth (PD = distance between the gingival margin and the bottom of the pocket, six points around each tooth), and bleeding on probing (BOP; six sites around each tooth) [32] after 30 s. For assessment of the severity of the familial burden of periodontitis, patients were asked about early tooth loss in first-degree relatives.

Periodontitis was classified according to the Centers for Disease Control and Prevention (CDC) [33] and in accordance with the consensus report of the 2017 World Workshop on the Classification of Periodontal and Peri-Implant Diseases and Conditions [34]. Severe (≥2 interproximal sites with CAL ≥ 6 mm (not on same tooth), ≥1 interproximal site with PD ≥ 5 mm), moderate (≥2 interproximal sites with CAL ≥ 4 mm (not on same tooth) or ≥2 interproximal sites with PD ≥ 5 mm (not on one tooth)), and mild/absent periodontitis (previously mentioned criteria did not apply) were documented.

Applying the new classification [34], the disease stages I to IV were defined by severity, complexity, extent, and distribution. Grading was not implemented in this study for ethical reasons (no X-rays were available, and there was no estimation of radiological bone loss).

#### 2.2.2. Three-Year Follow-Up Period

Between 2020 to 2021, 95 CV patients were included in the 3-year follow-up (drop-out rate of 6.9%). The follow-up was implemented as a telephone interview. If we were not successful in reaching the patient by telephone, civil registration offices were contacted and information about the current address or date of death were requested. The CV outcome after CABG surgery was evaluated using the major adverse cardiac and cerebrovascular events criteria (MACCE): 1. No event; 2. Myocardial infarction; 3. Low cardiac output syndrome; 4. Tachycardia (VT); 5. Angina pectoris; 6. Renewed revascularization surgery; 7. Cardiac decompensation; 8. Peripheral circulatory failure; 9. Stroke/transient ischemic attack (TIA)/prolonged reversible ischemic neurological deficit (PRIND); 10. Cardiac death; 11. Stroke death; and 12. Noncardiac death.

#### 2.2.3. Statistical Analysis

Statistical analyses of demographic and clinical data were carried out using the software SPSS v.25.0 package (IBM, Chicago, IL, USA). Values of *p* ≤ 0.05 were considered statistically significant. Continuous data was assessed for a normal distribution using the Kolmogorov–Smirnov test and the Shapiro–Wilk test. These data are reported as the median and 25th/75th interquartiles (non-normally distributed values). For statistical evaluation, the Kruskal–Wallis test was used. Categorical variables are documented as a percentage, applying the chi-squared test for statistical analyses.

### 2.3. Investigation of Subgingival Microbiota

The procedure for subgingival sample collection and DNA preparation and details of the sequencing of the microbiological organisms are reported in detail in Schulz et al., 2019 and 2021 [28,35].

#### 2.3.1. Subgingival Sampling

Subgingival samples were taken from the deepest pocket of each quadrant using sterile paper points (Hain Lifescience, Nehren, Germany). Subgingival samples of one individual were pooled. After drying the sample, the paper points were stored at −20 °C until DNA was isolated.

#### 2.3.2. DNA Isolation and Sample Preparation for Next-Generation Sequencing

The microbial DNA was extracted using a QIAamp^®^ DNA mini kit (Qiagen, Hilden, Germany) according to the manufacturer’s protocol. For next-generation sequencing, microbial DNA primers targeted the 16S V3 and V4 region were used (PCR conditions: 3 min 95 °C; 25 cycles: 30 s 95 °C, 30 s 55 °C, 30 s 72 °C; 5 min 72 °C; and held 4 °C; Mastercycler Gradient, Eppendorf AG, Hamburg, Germany). Subsequent DNA quality was assessed by applying a Bioanalyzer DNA 1000 chip. For DNA purification, Agencourt AMPure XP beads (Beckman Coulter, Brea, CA, USA) were used according to the manufacturer’s protocol. The Nextera XT Index Kit (Illumina, San Diego, CA, USA) was employed for dual indexing of different DNA samples (index PCR conditions: 3 min 95 °C; 8 cycles: 30 s 95 °C, 30 s 55 °C, 30 s 72 °C; 5 min 72 °C; and held 4 °C). After DNA purification and quality control, the DNA was evaluated using a Qubit^®^ dsDNA BR Assay Kit and Qubit^®^ Fluorometer (Thermofisher Scientific, Waltham, MA, USA) for library quantification and normalization. The different DNA samples were pooled (final concentration of 4 pM) and spiked with phiX control (4 pM). For sequencing analysis, MiSeq v3 reagents (Illumina, San Diego, CA, USA) were applied.

#### 2.3.3. Next-Generation Sequencing

The library was loaded onto an MiSeq (Illumina, San Diego, CA, USA) flow cell using a 600-cycle reagent cartridge and 2 × 301-bp paired-end sequencing. After the 65-h run, the FASTQ files were assessed using MiSeq Reporter software v2.6 (Illumina, San Diego, CA, USA).

#### 2.3.4. Data Analysis

Paired-end read assembly: The reads were merged using FLASH software [36]. Applying the QIIME workflow, the raw tags were preprocessed, and the quality filtered [37]. The tags were compared with a reference database for chimera depletion using the UCHIME algorithm [38].

Operational taxonomic unit (OTU) cluster and species annotation: For sequence analysis, Uparse software was employed [39]. Sequences showing >97% similarity were assigned to the same OTU.

For species annotation, we referred to the GreenGene Database based on the RDP classifier algorithm [40]. Normalization of the OTU abundance information was performed to a standard sequence corresponding to the sample with the least sequences. Output-normalized data were assessed for all subsequent analyses.

Alpha diversity: Alpha diversity was calculated with QIIME (V1.7.0) and assessed with R software (V2.15.3). To identify community diversity, the Shannon index was used (http://mothur.org.wiki/Shannon, accessed on 16 March 2022). Differences in the alpha diversity (Shannon indices) were assessed by the Wilcoxon signed-ranks test and corrected for multiple comparisons using a Benjamini–Hochberg false discovery rate (FDR) of 5%.

Beta diversity: Beta diversity was also calculated by applying the QIIME software (V1.7.0). The general distribution of the resulting bacterial community composition was evaluated using principal coordinates analysis (PCoA) with the R package ggplots2 (V2.15.3). The linear discriminant analysis (LDA) and effect size (LEfSe) pipeline was applied using the software Galaxy provided by Dr. Huttenhower (https://huttenhower.sph.harvard.edu/galaxy, accessed on 16 March 2022). Differences in the means among the two CV outcome groups (at all phylogenetic levels) were evaluated using the *t*-test, including FDR analysis.

## 3. Results

In this longitudinal cohort study, 102 consecutive CV patients requiring CABG surgery were included. After the 3-year follow-up, 95 patients remained in the study (drop-out rate: 6.9%). Of these 95 patients, 42 (44.2%) had suffered a secondary CV endpoint according to the applied MACCE criteria (angina pectoris (n = 14), cardiac decompensation (n = 7), ventricular tachycardia (n = 6), death due to a cardiac event (n = 5), renewed revascularization surgery (n = 4), stroke/TIA/PRIND (n = 3), peripheral circulatory failure (n = 2), and myocardial infarction (n = 1)). The anamnestic and clinical data are compared with respect to the CV outcomes in Table 1.

### 3.1. Microbiome Structure Analysis of the CV Patient Cohort

In total, 95 CV patients who had completed the 3-year follow-up were included in the analysis of subgingival microbiota. The microbial composition was evaluated by comparing CV patients with (44.2%) and without a secondary CV event according to the MACCE criteria (55.8%).

### 3.2. Relative Abundance in Relation to the Secondary CV Endpoint

In Figure 1, the species annotation results for the top ten most abundant phyla are displayed in bars with respect to the CV outcome (Figure 1). It is evident that *Firmicutes*, *Bacteroidota,* and *Fusobacteriota* represent the most abundant phyla in both groups (about 75% in each group). *Bacteroidota* represent the major component at 34.5% each. A distinction between the two groups, with and without a CV endpoint, at the phylum level is not evident. Similar results were obtained by comparing the two groups with regard to further taxonomic classifications (genus, family, order, and class; data available from the authors on request).

### 3.3. Alpha Diversity of Subgingival Microbiota Related to the Secondary CV Endpoint

Differences in the alpha diversity between the two CV groups were evaluated at the OTU level using the Shannon index (Figure 2). Despite showing no statistically significant differences in the Shannon indices, the alpha diversity was slightly lower in patients with secondary CV endpoints than in patients without CV endpoints (*p* = 0.49).

### 3.4. Beta Diversity of Subgingival Microbiota Considering the Secondary CV Endpoint

PCoA, LDA, LEfSe, and *t*-tests were used to assess possible differences in the beta diversity of the microbial composition considering the CV outcomes.

No general differentiation between the patient groups according to the CV endpoint was observed in the PCoA (data available from the authors on request). LDA and LEfSe were performed to demonstrate the distribution of the relative abundances between the two patient groups (Figure 3). We applied these models to rank the bacterial taxonomic categories based on the relative differences between the study groups. It showed that species of the genus *Campylobacter* were more characteristic for the risk of occurrence of a CV endpoint. This link was evident across all taxonomic categories up to the phylum Campylobacterota. This correlation was also confirmed in the *t*-test, taking multiple testing into account (including a Benjamin–Hochberg false discovery rate of 5%; *p* = 0.015; Figure 4). In contrast, patients without a secondary CV event more commonly exhibited bacteria of the family Prevotellaceae (Figure 3).

## 4. Discussion

Due to the high morbidity and mortality associated with CV diseases, investigation of disease-contributing factors represents an important social and socioeconomic priority. A variety of chronic infectious, inflammatory, and immune diseases have also been associated with a higher risk of CV disease and its sequelae [9]. Among these diseases, chronic inflammatory periodontitis is of great importance [9]. In this context, the oral microbiota in combination with the host immune response has shifted into the focus of clinical research [25]. Here, attention is being paid not only to the connection between the oral microbiota and the manifestation of CV diseases [7] but also to its impact on recurrent CV events [20]. However, the primary focus of these studies, especially when considering CV prognosis, has been on individual periodontal pathogenic species [20,41]. For this reason, the complex microbiota were evaluated as a CV prognostic factor in this study. Innovative analytical methods such as next-generation sequencing of 16S rRNA genes can evaluate ecological changes that consider the complexity of the oral microbiota (polymicrobial synergy and dysbiosis model) [26,27]. We used this strategy based on the V3/V4 hypervariable regions, which offers advantages regarding the sequencing depth, error rates, costs, and comparability with other studies [42].

### 4.1. Relative Abundance in Relation to the Secondary CV Endpoint

In a follow-up period of 1 year, it already became apparent that a bacterium (Saccharibacteria phylum; class: TM7-3, order: CW040, family: F16) could have an impact on the CV prognosis in survival analyses [28]. In the continuation of this study, we wanted to verify these results in a follow-up period of 3 years. In terms of the relative abundance at the phylum level, we did not observe any differences (top ten phyla) in relation to the combined CV endpoint (Figure 1). *Firmicutes*, *Bacteroidetes*, and *Fusobacteria* formed the largest proportion, with a combined 75%. Similar distributions were also found in other study populations comprising periodontitis patients and patients suffering from diabetes [43].

### 4.2. Alpha Diversity of Subgingival Microbiota Related to the Secondary CV Endpoint

When investigating possible changes in the composition of the microbiota as a function of the CV outcomes, we did not find any remarkable differences in the alpha diversity (Shannon index). As shown in the 1-year follow-up, the 3-year follow-up confirmed that patients with a combined CV endpoint exhibited decreased alpha diversity [28] (Figure 2). It can be hypothesized that this decrease in richness and evenness could indicate a higher level of microbial uniformity in CV patients with adverse outcomes. Although a change in the Shannon index in regard to the oral microbiota has not yet been shown to be related to CV outcome, correlations with other diseases have been demonstrated [35,44,45]. Furthermore, in investigating a possible connection between microbial colonization in this particular intestinal microbiota and CV disease, a decrease in the Shannon index was also shown in relation to CV disease [46,47].

### 4.3. Beta Diversity of Subgingival Microbiota Considering the Secondary CV Endpoint

Microbial markers related to a combined CV endpoint were identified by applying LDA and LEfSe analysis (Figure 3). It was striking that bacteria of the genus *Campylobacter* (and all taxonomic categories above it) were most strongly associated with the adverse CV outcome. This association was proven in *t*-tests applying a Benjamin–Hochberg false discovery rate of 5% in order to account for the variety of microbial species (*p* < 0.05, for all taxonomic levels from the phylum to the genus *Campylobacter*). The genus *Campylobacter* comprises 32 species and 9 subspecies [48] and includes a number of bacteria that have been linked to human disease, including gastrointestinal infections (e.g., gastroenteritis, ulcerative colitis, and Crohn’s disease) and extragastrointestinal infections (e.g., meningitis, bronchial abscess, bacteremia, and perimyocarditis) [49]. So far, two *Campylobacter* species, *C. rectus* and *C. gracilis*, have been associated with periodontal disease [48,50]. In addition, *C. rectus* was more frequently detected in both subgingival plaque and atherosclerotic plaque in patients with CV disease [51,52]. Furthermore, elevated levels of serum IgA/IgG antibodies for *C. rectus* were associated with periodontitis and acute coronary syndrome [53]. These studies may also suggest that species of the genus *Campylobacter* form a link between periodontal and CV diseases.

However, our study is the first to demonstrate that subgingival occurrence of bacteria of the genus *Campylobacter* can be considered as a predictor of future adverse CV events.

### 4.4. Limitations of the Study

The present study was performed as a longitudinal cohort study. It was conducted to provide hypotheses on the possible prognostic value of periodontal species and adverse CV events within a 3-year follow-up period.

The MACCE criteria were used as the basis for assessing a CV endpoint. For this purpose, the data were obtained from medical records, from patients or their relatives (standardized questionnaire, telephone interview), or from civil registry offices. In this respect, false information from patients or their relatives due to possible personal misinterpretations of their health status cannot be excluded.

The results refer to secondary but not to primary CV prevention due to the chosen study design. It could be shown that species of the genus *Campylobacter* are linked to an adverse CV outcome only in patients with a CV predisposition (at least 60% stenosis of one of the main coronary arteries). Therefore, oral colonization with bacteria of this genus cannot be considered a primary CV risk factor in the normal population.

Additionally, only the bacterial DNA was examined by means of high-throughput DNA sequencing analysis (next-generation sequencing). Therefore, conclusions about the bacterial vitality and the current infection status are not possible due to the study design.

## 5. Conclusions

In the present study, species of the genus *Campylobacter* were identified as prognostic factors for adverse CV events at a 3-year follow-up in patients undergoing coronary bypass surgery. The results highlight the implication of the complex consideration of the subgingival microbiota as a predictive factor for adverse CV events.

## Figures and Tables

**Figure 1 biomedicines-10-01801-f001:**
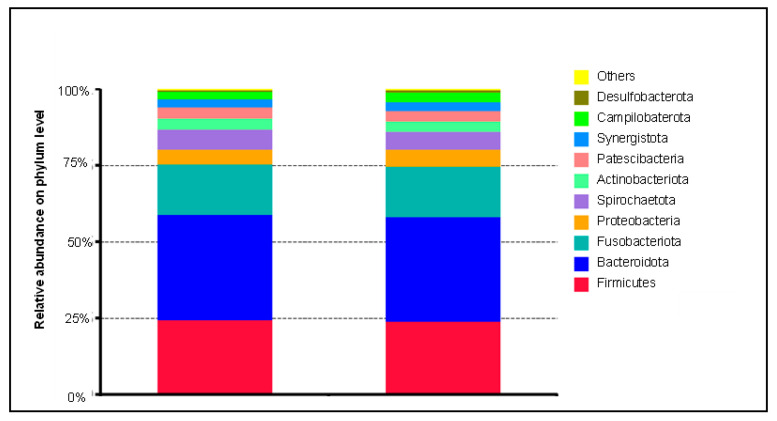
Microbial community composition at the phylum level comparing CV patients without and with a combined CV endpoint according to the MACCE criteria in the 3-year follow-up.

**Figure 2 biomedicines-10-01801-f002:**
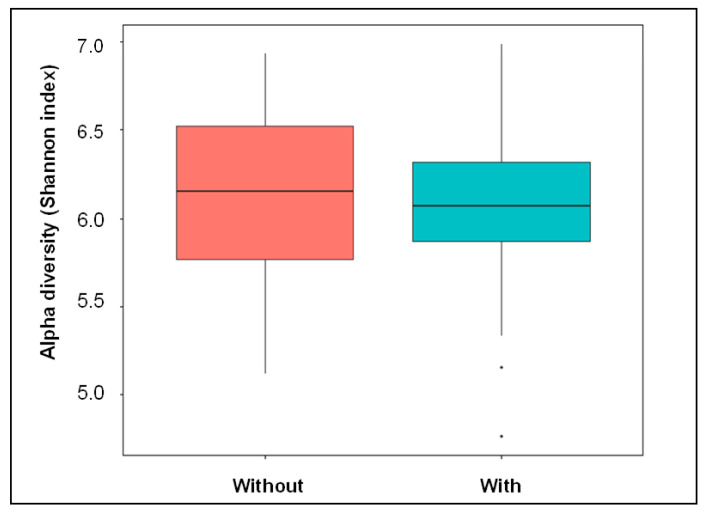
Comparison of the alpha diversity of patients without and with a combined cardiovascular endpoint event using Shannon index applied at the OTU level.

**Figure 3 biomedicines-10-01801-f003:**
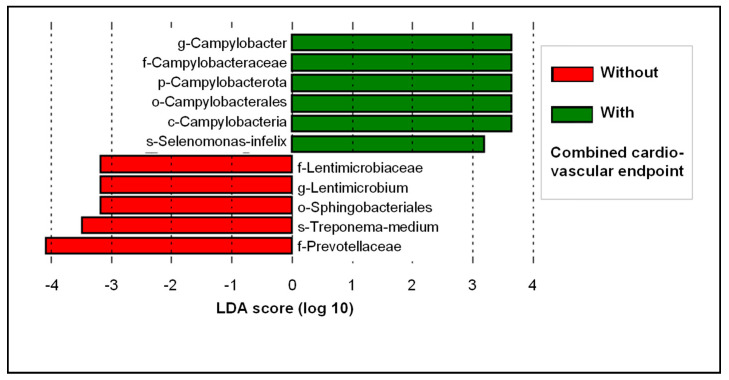
Linear discriminant analysis (LDA) and effect size analysis (LEfSe) considering the combined cardiovascular endpoint according to the MACCE criteria in the 3-year follow-up.

**Figure 4 biomedicines-10-01801-f004:**
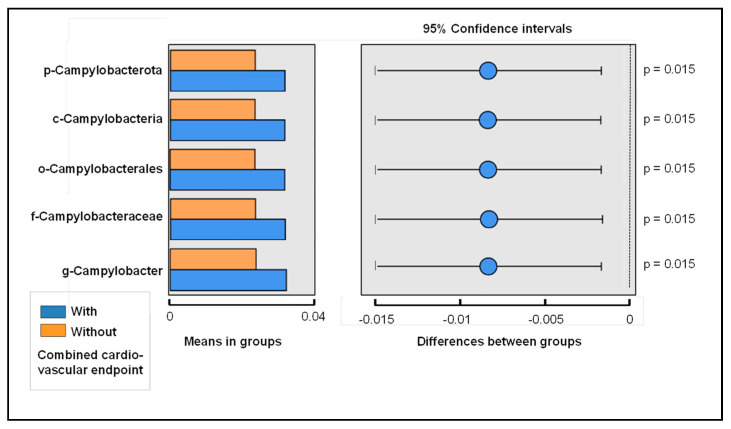
Differences in means comparing patients with and without a combined cardiovascular endpoint according to the MACCE criteria in the 3-year follow-up (*t*-test, including a Benjamin–Hochberg false discovery rate of 5%).

**Table 1 biomedicines-10-01801-t001:** Demographic and clinical characteristics (including demographical and anamnestic data, history of previous diseases, and periodontal status) considering the 3-year cardiovascular outcome (combined endpoint refers to the occurrence of major adverse cardiac and cerebrovascular events criteria: 1. No event; 2. Myocardial infarction; 3. Low cardiac output syndrome; 4. Ventricular tachycardia; 5. Angina pectoris; 6. Renewed revascularization surgery; 7. Cardiac decompensation; 8. Peripheral circulatory failure; 9. Stroke/Transient ischemic attack/Prolonged reversible ischemic neurological deficit; 10. Cardiac death; and 11. Stroke death) (PTCA, percutaneous transluminal coronary angioplasty; DMF/T, decayed missing filled/teeth).

Variable	Without Combined Endpoint (n = 53)	With Combined Endpoint (n = 42)	*p*-Value
**Demographic data**			
Age (years) (Median (25–75th IQR)	68.0 (60.0–74.0)	72.0 (61.0/76.25)	0.06
Female gender (%)	11.3	11.9	1
Body mass index (kg/m^2^) (Median (25–75th IQR)	28.7 (25.9/31.1)	29.0(25.3/31.1)	0.805
**Smoking**			
Current (%)	20.8	21.4	0.755
Past (%)	45.3	38.1	
Never (%)	34	40.5	
**History of**			
Diabetes mellitus (%)	37.7	45.2	0.597
Hypertension (%)	84.9	92.9	0.379
Dyslipoproteinemia (%)	88.7	73.8	0.108
Peripheral arterial disease (%)	7.5	28.6	0.015
Coronary heart disease (%)	34	40.5	0.66
Myocardial infarction (%)	20.8	38.1	0.103
Stroke/transient ischemic attack (%)	9.4	9.5	1
Angina pectoris (%)	75.5	71.4	0.834
PTCA/stent (%)	9.4	16.7	0.458
**Periodontitis classification (CDC)**			
None or mild (%)	0	0	0.912
Moderate (%)	35.8	21.4	
Severe (%)	64.2	78.6	
**Staging**			
1 (%)	0	0	0.493
2 (%)	1.9	0	
3 (%)	83	88.1	
4 (%)	15.1	11.9	
**Periodontal and dental parameter**			
Plaque index (%) (Median (25–75th IQR)	1.25 (0.9/1.7)	1.25 (0.8/1.69)	0.629
Bleeding index (%) (Median (25–75th IQR)	18.4 (12.0/33.1)	16.9 (8.6/33.3)	0.436
Pocket depth (mm) (Median (25–75th IQR)	2.9 (2.5/3.5)	3.0 (2.6/3.5)	0.481
Attachment loss (mm) (Median (25–75th IQR)	3.8 (3.2/4.8)	4.2 (3.2/4.9)	0.462
DMF/T (n) (Median (25–75th IQR)	18.0 (14.0/23.0)	17.0 (12.75/22.0)	0.462
Missing teeth (n) (Median (25–75th IQR)	6.0 (3.5/16.0)	6.0 (3.0/12.25)	0.761

## Data Availability

The comprehensive study protocol is available from the authors. Data can be provided by the authors upon request.

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
