# Peer review of "Campylobacter Species of the Oral Microbiota as Prognostic Factor for Cardiovascular Outcome after Coronary Artery Bypass Grafting Surgery"

_biomedicines, 2022, doi:10.3390/biomedicines10081801_

Round 1
Reviewer 1 Report
The paper presented to me for review (Campylobacter species of the oral microbiota as prognostic factor for cardiovascular outcome after coronary artery bypass grafting surgery) concerns the medically important matter of bacterial infections and their correlation with cardiovascular disease. The authors analysed 102 cases of patients. In my opinion, this is not an impressive group but sufficient to draw statistical conclusions. The follow-up time of the patients was 3 years.
The methodologies used in the preparation of the paper were clearly and comprehensively presented.
In their work, the authors used statistical methods that were adequate to their purpose. They presented their results in the form of 1 table and 4 figures.
The results of the study were clearly and transparently presented.
The authors cited 55 items of current and relevant literature in their work
In my opinion, the work deserves to be accepted for publication without significant revisions
Author Response
On behalf of all authors, I would like to thank the reviewer very much for his favorable assessment.
Reviewer 2 Report
Could the authors keep just the citation and pu the links in the bibliography?Paired-End Read Assembly: The reads were merged using FLASH (V1.2.7, 190 http://ccb.jhu.edu/software/FLASH) [36]. Applying the QIIME workflow the raw tags 191 were preprocessed and the quality filtered (V1.7.0, http://qiime.org/index.html) [37]. The 192 tags were compared with a reference database for chimera depletion using the UCHIME 193 algorithm (http://www.drive5.com/usearch/manual/uchime_algo.html); 194 http://drive5.com/uchime/uchime_download.html (accessed on 07.05.2021)) [38
Also for this - The linear discriminant analysis effect size (LEfSe) pipeline was applied accord- 213 ing to Galaxy (https://huttenhower.sph.harvard.edu/galaxy/), https://hut- 214 tenhower.sph.harvard.edu/galaxy/datasets/7c8874d23329d3dd/display/?preview=True). 215 Differences in means among the two CV outcome groups (at all phylogenetic levels) were 216 evaluated using t-test, including FDR analysi-
Author Response
Dear Reviewer,
Thank you very much for your invaluable comments while revising our manuscript. We introduced your comments and critical remarks in the revised manuscript.
Here, we want to explain how the manuscript was revised:
Could the authors keep just the citation and put the links in the bibliography?
All links addressed have been removed from the main text and listed in the bibliography.
Since there is no citation for the calculation of the LefSe using Galaxy software, but rather it is a website that is made available by Dr. Huttenhower for anyone to use, we have revised the sentence as follows:
“The linear discriminant analysis effect size (LEfSe) pipeline was applied using the software Galaxy provided by Dr. Huttenhower (https://huttenhower.sph.harvard.edu/galaxy).”
We hope we have satisfied the concerns of the reviewer with regard to the points marked.
With kindest regards
Susanne Schulz on behalf of all authors
Reviewer 3 Report
The current manuscript presents an original study that aimed to offer a potential prognostic connection between the oral microbiota and cardiovascular events in patients with coronary bypass surgery.
Introduction adequately presents the background of the study, but needs a few minor modifications: rows 74-76 - the word "however" is used too much; lines 78-89 must be rewritten in order to better state the aim as well as the originality and added value of the study.
Material and methods: the study design is adequate and the methods are well described, with enough details in order to be well understood and easy to reproduce.
Results are well described, statistical analysis is adequate and relevant, the tables and figures as well as their legends are suitable.
Discussions are very well conducted and point out the novelty of this study in comparison with other related papers. Presenting the limitations of the study is also very welcome.
Conclusion is in accordance with the results.
The authors should revise the manuscript for English language mistakes, it should be edited by a proficient speaker.
Overall, I recommend publication of the manuscript after minor corrections.
Author Response
Dear Reviewer,
Thank you very much for your invaluable comments while revising our manuscript. We introduced your comments and critical remarks in the revised manuscript.
Here, we want to explain how the manuscript was revised. The changes are indicated using blue color.
Minor corrections of the introduction
The use of the word “however” was revised in rows 74-76.
The lines 78-89 were rewritten in order to better state the aim and emphasize the value and the originality of the study.
The authors should revise the manuscript for English language mistakes; it should be edited by a proficient speaker.
The manuscript was proofread for English by native speaker Sherryl Sundell, who was managing editor of the International Journal of Cancer and who has many years of experience as a professional editor.
We hope we have satisfied the concerns of the reviewer with regard to the points marked.
With kindest regards
Susanne Schulz on behalf of all authors